# ADVANCING MULTIMODAL UNIFIED DISCRETE REPRESENTATIONS

## ABSTRACT

To enhance the interpretability of multimodal unified representations, many studies have focused on discrete unified representations. These efforts typically start with contrastive learning and gradually extend to the disentanglement of modal information, achieving solid multimodal discrete unified representations. However, existing research often overlooks two critical issues: **1)** Different modalities have unique characteristics, and a uniform alignment approach does not fully exploit these traits; **2)** The use of Euclidean distance for quantization in discrete representations often overlooks the important distinctions among different dimensions of features, resulting in redundant representations after quantization. To address these issues, we propose Fine and Coarse Cross-modal Information Disentangling (FCCID) and Training-Free Optimization of Codebook (TOC). These methods respectively perform fine and coarse disentanglement of information based on the specific characteristics of different modalities and refine the unified discrete representations obtained from pretraining. Compared to the previous state-of-the-art, our model demonstrates significant performance improvements. The code is provided in the supplementary materials.

## 1 INTRODUCTION

Humans' capacity to integrate multimodal information, such as text, audio, and visual, has inspired research on extracting unified information from multimodal data (Harwath et al., 2018; Miech et al., 2019; Shvetsova et al., 2022; Monfort et al., 2021). Researchers aim to develop models that learn unified representations across modalities, using techniques like contrastive learning to map semantically similar multimodal data closer in the embedding space (Radford et al., 2021; Luo et al., 2022; Xu et al., 2021), achieving notable results in downstream tasks like zero-shot cross-modal retrieval. However, the unbounded nature of the continuous embedding space poses challenges in interpretability. To address this, recent works have explored constructing discrete embedding spaces with prototypes or codebooks, enhancing cross-modal learning and model interpretability (Duan et al., 2022; Liu et al., 2021a; Lu et al., 2022; Zhao et al., 2022; Xia et al., 2024).

While recent works has demonstrated incredible achievements in multimodal unified representation, there are limitations in terms of the efficiency of embedding space utilization and the granularity of alignment. **1)** In the unified discrete representation of multimodal data, some studies focus on coarse-grained semantic alignment (Duan et al., 2022), others on fine-grained alignment (Xia et al., 2024), and yet others consider both fine and coarse alignments simultaneously (Liu et al., 2021a). However, these approaches align text with audiovisual data in the same granularity, overlooking the inherent differences between modalities: audiovisual data have temporal fine-grained connections, whereas text represents holistic semantics. **2)** Previous work, whether through contrastive learning (Liu et al., 2021a), teacher-student distillation (Duan et al., 2022), or information disentanglement (Xia et al., 2024), has aimed to achieve a multimodal discrete unified representation that retains shared information across modalities. However, this shared information still contains redundant background elements that do not contribute to the core semantics. We propose that refining the modal-general features from this perspective could lead to improvements. According to previous work (Breiman, 2001; Wojtas & Chen, 2020), the significance of features varies across different dimensions, and the selection of appropriate dimensions can accelerate inference speed and enhance model performance. However, preceding efforts (Liu et al., 2021a; Lu et al., 2022; Zhao et al., 2022; Xia et al., 2024) overlook this issue due to the inherent constraints of the codebook and

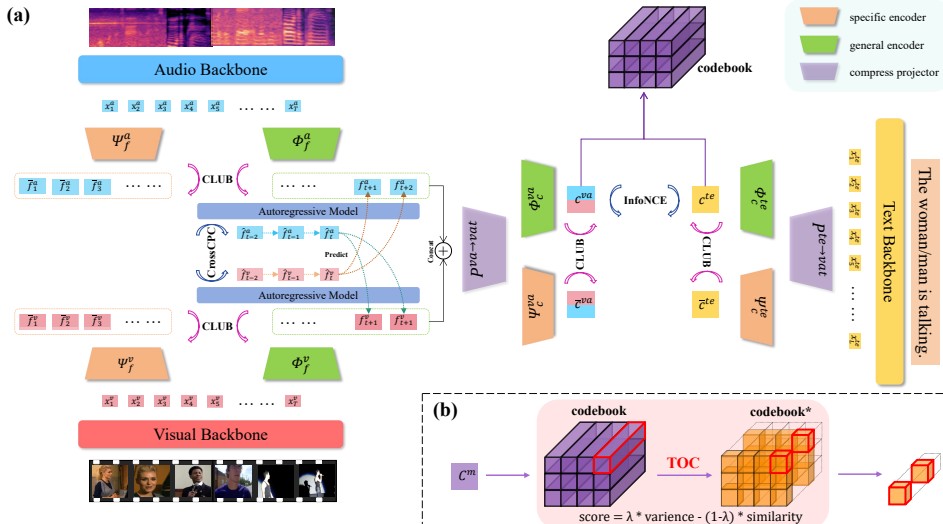

Figure 1: (a) The FCCID encoder architecture. On the left, audio and video undergo fine-grained mutual information separation and alignment using modal-general encoders $\Phi_f^a, \Phi_f^v$ and modal-specific encoders $\Psi_f^a, \Psi_f^v$. The CLUB module separates specific information $\overline{\mathbf{f}}_i^a, \overline{\mathbf{f}}_i^v$ from general information $\mathbf{f}_i^a, \mathbf{f}_i^v$, while CrossCPC aligns the general information across modalities. This is followed by compressing the features into unified audiovisual representations. On the right, coarse-grained mutual information separation and alignment are conducted with audiovisual data and text, resulting in a unified discrete representation across all three modalities. (b) An example of TOC, where the refinement of the codebook requires computation **only once**. The dimension selections do not require repeated calculations, and the entire process is training-free.

the quantization method based on Euclidean distance. This approach treats all feature dimensions equally without considering the importance of individual dimensions, resulting in interference from irrelevant dimension information.

To address the aforementioned issues, we propose the FCCID approach tailored to modal differences, as shown in Figure 1(a). We first perform fine-grained alignment and disentanglement of audiovisual data, followed by compression. Then, the compressed holistic semantics are aligned and disentangled with text in a coarse-grained manner. This method effectively preserves the temporal knowledge of audiovisual data while ensuring semantic alignment across the three modalities. And inspired by a training-free adapter (Zhang et al., 2022; Zhu et al., 2023) and drawing on the concept of feature importance (Breiman, 2001; Wojtas & Chen, 2020; Xue et al., 2022; Zhu et al., 2023), we propose the TOC that starts from a pre-trained unified discrete representation space and optimizes it without additional training. As shown in Figure 1(b), this innovative approach assesses the importance of feature dimensions in the pre-trained model's codebook without requiring further training. By refining the dimensions of the codebook, TOC reduces training parameters and time while improving experimental outcomes.

The main contributions of this work are summarized as follows:

- We introduce **FCCID**, which disentangles information based on the distinct characteristics of text and audiovisual modalities, effectively preserving the temporal information of audiovisual data along with the overarching semantic information across all three modalities. Furthermore, it efficiently extracts the general information shared across modalities through disentanglement.

- We propose the **TOC**, a novel approach that precisely identifies the importance of feature dimensions through calculations without the need for additional training. To the best of our knowledge, this is the first attempt at training-free optimization of the codebook, applicable to both multimodal unified codebook and single-modal codebook. This method is promising and easily transferable.

- Our method significantly outperformed state of the art (SOTA), across various tasks in the cross-modal generalization setup, showcasing its effectiveness in multimodal learning. Specifically, FCCID, TOC, and their combination outperformed before SOTA by 2.16%, 1.06%, and 2.96% respectively, on four downstream tasks. Furthermore, the study demonstrates the model's potential applications in a broader range of retrieval, generation and reconstruction tasks.

## 2 RELATED WORK

In this section, we will introduce recent works on multi-modal unified representations and their distinctions, as well as explorations of training-free methods in other fields. For specific details, please refer to Appendix A.

## 3 METHOD

In this section, we introduce the proposed FCCID and TOC, aimed at effectively enhancing the capability of multimodal unified representations. In Sectio 3.1, we elaborate on the foundational principles and design rationale behind the FCCID. In Section 3.2, we introduce the two internal code metrics that constitute TOC.

### 3.1 FINE AND COARSE CROSS-MODAL INFORMATION DISENTANGLING

FCCID addresses the inherent differences between text and audiovisual modalities with a two-step process for information disentanglement and alignment, namely Fine Cross-modal Information Disentangling (FCID) and Coarse Cross-modal Information Disentangling (CCID). FCID finely extracts the modal-general information shared between audio and video, while CCID further disentangles and aligns this information with text at a coarser granularity. This approach not only preserves the temporal fine-grained details of audiovisual data but also establishes a unified representation space that incorporates common information across all three modalities.

### 3.1.1 FINE CROSS-MODAL INFORMATION DISENTANGLING

Given paired audio-video modalities, $(\mathbf{x}_i^a, \mathbf{x}_i^v)_{i=1}^N$, we utilize two fine modal-general encoders, $\Phi_f^a$ and $\Phi_f^v$, to extract fine modal-general features $\mathbf{f}_i^a$ and $\mathbf{f}_i^v \in \mathbb{R}^{T \times D}$, and employ two fine modal-specific encoders, $\Psi_f^a$ and $\Psi_f^v$, to obtain fine modal-specific features $\overline{\mathbf{f}}_i^a$ and $\overline{\mathbf{f}}_i^v \in \mathbb{R}^{T \times D}$ from the audio and video modalities, respectively. Here, $N$, $T$, and $D$ represent the number of samples, the length of audio-video sequences, and the feature dimension, respectively:

$$\mathbf{f}_i^m = \Phi^m(\mathbf{x}_i^m), \ \overline{\mathbf{f}}_i^m = \Psi^m(\mathbf{x}_i^m), \ m \in \{a, v\}. \tag{1}$$

Subsequently, we utilize CLUB (Cheng et al., 2020) to minimize the mutual information between the fine modal-specific features $\mathbf{f}_i^m$ and $\overline{\mathbf{f}}_i^m$. At the same time, we apply Cross-modal Contrastive Predictive Coding (CrossCPC) (Oord et al., 2018) to maximize the mutual information between $\mathbf{f}_i^m$ and $\mathbf{f}_i^n$. The details of this approach are outlined below:

**Mutual Information Minimization:** CLUB (Cheng et al., 2020) could optimize the mutual information upper bound, demonstrating superior advantages in information disentanglement. Given two variables $\mathbf{x}$ and $\mathbf{y}$, the objective function of CLUB is defined as:

$$I_{vCLUB}(\mathbf{x}; \mathbf{y}) := \mathbb{E}_{p(\mathbf{x}, \mathbf{y})}[\log q_\theta(\mathbf{y}|\mathbf{x})] - \mathbb{E}_{p(\mathbf{x})}\mathbb{E}_{p(\mathbf{y})}[\log q_\theta(\mathbf{y}|\mathbf{x})]. \tag{2}$$

We use CLUB to optimize the mutual information upper bound between fine modal-general features $\mathbf{f}_i^m$ and fine modal-specific features $\overline{\mathbf{f}}_i^m$, where $q_\theta$ is the variational approximation of ground-truth posterior of $\mathbf{y}$ given $\mathbf{x}$ and can be parameterized by a network $\theta$.

$$\hat{I}_{vCLUB_f} = \frac{1}{N}\sum_{i=1}^{N}[\frac{1}{T}\sum_{t=1}^{T}\log q_\theta(\bar{\mathbf{f}}_i^m|\mathbf{f}_i^m) - \frac{1}{N}\frac{1}{T}\sum_{j=1}^{N}\sum_{t=1}^{T}\log q_\theta(\bar{\mathbf{f}}_j^m|\mathbf{f}_i^m)], \; m \in \{a, v\}. \quad (3)$$

**Mutual Information Maximization:** Contrastive Predictive Coding (CPC) (Oord et al., 2018) aims to maximize the mutual information between sequence items by predicting future samples using autoregressive models and is widely adopted in self-supervised learning. Humans possess the ability to not only predict subsequent scenarios from a current modality but also to associate potential situations in other modalities, such as inferring forthcoming audio from video or text or envisioning subsequent scenes from audio. Given fine general features $\mathbf{f}^a, \mathbf{f}^v \in \mathbb{R}^{T \times D}$, a prediction horizon of R steps, and a random time moment $t \in (0, \text{T-R}]$, two single-layer unidirectional LSTMs are used to summarize the information of all $\mathbf{f}_{\leq t}^a, \mathbf{f}_{\leq t}^v$, yielding three context representations as $\mathbf{o}_t^m = \text{LSTM}(\mathbf{f}_{\leq t}^m)$.

For modality M, we first select a set $Z_{neg}$ of N-1 random negative samples and one positive sample $\mathbf{f}_{t+r}^n$ from modality N, then use $\mathbf{o}_t^m$ to predict r-th future step $\mathbf{f}_{t+r}^n$ in modality N, and the loss for all modality can be optimized as:

$$L_{cpc}^{m2n} = -\frac{1}{R}\sum_{r=1}^{R}\log\left[\frac{\exp\left(\mathbf{f}_{t+r}^n W_r^m \mathbf{o}_t^m\right)}{\sum_{\mathbf{f}_j \in Z_{neg}}\exp\left(\mathbf{f}_j^n W_r^m \mathbf{o}_t^m\right)}\right], \; m, n \in \{a, v\}. \quad (4)$$

### 3.1.2 COARSE CROSS-MODAL INFORMATION DISENTANGLING

CCID initially sets up two projections, $P^{te \to vat}$ for compressing textual features and $P^{va \to vat}$ for compressing the audiovisual modal-general features obtained from FCID. Subsequently, it configures two coarse modal-specific encoders, $\Psi_c^{av}$ and $\Psi_c^{te} \in \mathbb{R}^D$, to extract coarse modal-specific features $\bar{\mathbf{c}}_i^{av}$ and $\bar{\mathbf{c}}_i^{te}$, and two coarse modal-general encoders, $\Phi_c^{av}$ and $\Phi_c^{te}$, are employed to derive coarse modal-general features $\mathbf{c}_i^{av}$ and $\mathbf{c}_i^{te} \in \mathbb{R}^D$ from the audiovisual and textual modalities, respectively:

$$\mathbf{c}_i^m = \Phi_c^m(P^{m \to vat}(\mathbf{x}_i^m)), \; \bar{\mathbf{c}}_i^m = \Psi_c^m(P^{m \to vat}(\mathbf{x}_i^m)), \; m \in \{av, te\}. \quad (5)$$

The subsequent process of information disentanglement and alignment is similar to that of FCID, where the CLUB (Cheng et al., 2020) method continues to be used for minimizing mutual information, while the method for maximizing mutual information has been switched to InfoNCE (Oord et al., 2018).

**Mutual Information Minimization:** We use CLUB to optimize the mutual information upper bound between coarse modal-general features $\mathbf{c}_i^m$ and fine modal-specific features $\bar{\mathbf{c}}_i^m$, $m \in \{av, te\}$, similar to $\hat{I}_{vCLUB_f}$ in FCID:

$$\hat{I}_{vCLUB_c} = \frac{1}{N}\sum_{i=1}^{N}[\frac{1}{T}\sum_{t=1}^{T}\log q_\theta(\bar{\mathbf{c}}_i^m|\mathbf{c}_i^m) - \frac{1}{N}\frac{1}{T}\sum_{j=1}^{N}\sum_{t=1}^{T}\log q_\theta(\bar{\mathbf{c}}_j^m|\mathbf{c}_i^m)], \; m \in \{av, te\}. \quad (6)$$

**Mutual Information Maximization:** Since the coarse information lacks a sequential structure, we transitioned the contrastive learning approach from CPC to InfoNCE, as described below:

$$L_{nce} = -\frac{1}{N}\sum_{i=1}^{N}\log\left[\frac{\exp(\text{sim}(c_i^m, c_i^n)/\tau)}{\sum_{j=1}^{N}\exp(\text{sim}(c_i^m, c_j^n)/\tau)}\right], \; m, n \in \{av, te\}. \quad (7)$$

Then we use the codebook to further help the final alignment of the three modalities into a unified discrete space, the latent codebook $\mathbf{e} \in R^{H \times D}$ is shared across modalities audio, video, and text, where $T, H, D$ represent time, size of the discrete latent space, and hidden dimension, respectively.

Apply vector quantized operation to map coarse model-general feature $\mathbf{f}_i^{av}, \mathbf{f}_i^{te}$ to discrete latent codes, $t \in [0, T)$:

$$\hat{\mathbf{c}}_{i,t}^m = VQ(\Phi_c^m(\mathbf{x}_i^m)) = VQ(\mathbf{c}_{i,t}^m) = e_l,$$
$$\text{where } l = argmin_j ||\Phi_c(x) - e_j||_2, \ m \in \{av, te\}. \tag{8}$$

Then, we combine $\hat{\mathbf{c}}_i^m$ with $\bar{\mathbf{c}}_i^m$ together to reconstruct original features:

$$\underbrace{||\mathbf{x}_i^m - D(\hat{\mathbf{c}}_i^m; \bar{\mathbf{c}}_i^m)||_2^2}_{\text{reconstruction loss}} + \underbrace{||\text{sg}[\phi_k^m(\mathbf{x}_i^m)] - \mathbf{e}||_2^2}_{\text{VQ loss}} + \underbrace{\beta||\phi_k^m(\mathbf{x}_i^m) - \text{sg}[\mathbf{e}]||_2^2}_{\text{commitment loss}}, \quad m \in \{av, te\} \tag{9}$$

where $\beta$ is set to 0.25, and sg denotes the stop gradient operation. We employ the Exponential Moving Average (EMA) strategy to replace the Vector Quantization (VQ) loss. The reconstruction loss ensures that the compressed latent codes $e_l$ retain the general information of different modalities. Ideally, $\mathbf{z}_i^a$, $\mathbf{z}_i^b$, and $\mathbf{z}_i^c$, encoded from different modalities with the same semantics, should be mapped to the same discrete latent code. However, in the absence of effective supervision, the presence of a modality gap may lead to $\mathbf{z}_i^a$, $\mathbf{z}_i^b$, and $\mathbf{z}_i^c$ converging to distinct regions of the codebook (Zhao et al., 2022; Liu et al., 2021a). Consequently, we need to minimize the mutual information between the general result and the specific result, as well as maximize the mutual information among the general results of different modalities.

The overall objective of FCCID is a combination of these loss functions across both layers:

$$L = L_{\text{recon}} + L_{\text{commit}} + L_{\text{cmcm}} + L_{\text{MImax}} + L_{\text{MImin}}, \tag{10}$$

where $L_{\text{recon}}$ is the reconstruction loss that merges the modal-specific and modal-general results for each modality and compares them with the original input using MSE loss, $L_{\text{commit}}$ is the commitment loss that computes the MSE loss between the modal-general results and their quantized codes, $L_{\text{cmcm}}$ is the objective loss proposed by Liu et al. (2021a), which also promotes the alignment among modalities, $L_{\text{MImax}} = L_{\text{cpc}} + L_{\text{nce}}$ is the loss that enhances cross-modal alignment and inference by predicting future samples in one modality using information from another, and $L_{\text{MImin}} = \hat{I}_{vCLUB_f} + \hat{I}_{vCLUB_c}$ represents the mutual information loss concerning the modal-specific and modal-general results within each modality.

## 3.2 TRAINING-FREE OPTIMIZATION CODEBOOK

Discrete unified representation spaces commonly employ a codebook structure, where modalities are updated based on the Euclidean distance between their features and the codebook codes. This dimension-equal-weighted update strategy does not consider the varying importance of feature dimensions, leading to redundancy in the final discrete space. Therefore, we propose two metrics, Inter-Code Similarity and Inter-Code Variance, to refine the information in the unified space. Notably, our approach, TOC, focuses on optimizing the pre-trained codebook and performs calculations independently of downstream information, distinguishing it from methods like APE (Zhu et al., 2023), which rely on few-shot data to determine the most relevant feature dimensions. Additionally, TOC is designed to tackle more complex downstream tasks, while APE is primarily constrained to image classification.

**Inter-code Similarity:** This metric aims to enhance the distinctiveness of codes by extracting feature dimensions that minimize code similarity. We represent the unified representation codebook of modalities as $\mathbf{e} \in \mathbb{R}^{H \times D}$, where $H, D$ denote the size of the discrete latent space and hidden dimension, respectively.

Assuming the existence of a classification dataset with $C$ categories, acquiring its complete data enables the calculation of the average similarity, denoted as $S$. In an open-world setting, we may assume that the prior probabilities of all categories are equal, denoted as $\frac{1}{C}$. We adopt cosine similarity, $\delta(\cdot, \cdot)$, as the chosen metric:

$$S = \frac{1}{C^2} \sum_{i=1}^{C} \sum_{\substack{j=1 \\ j \neq i}}^{C} \frac{1}{N^i N^j} \sum_{m=1}^{N^i} \sum_{n=1}^{N^j} \delta(\mathbf{x}^{i,m}, \mathbf{x}^{j,n}), \tag{11}$$

where $\mathbf{x}^{i,m}$ and $\mathbf{x}^{j,n}$ denote the input features for the $m$-th and $n$-th samples of categories $i$ and $j$, respectively. $N^i$ and $N^j$ represent their respective total number of training samples.

Each code in the codebook, $\mathbf{e}^i \in \mathbb{R}^D, i \in [0, L)$, can be considered as a distinct semantic cluster center, representing a category. Therefore, we can simplify the average similarity calculation by considering each code as representing one category:

$$S = \frac{1}{L^2} \sum_{i=1}^{L} \sum_{\substack{j=1 \\ j \neq i}}^{L} \delta(\mathbf{e}^i, \mathbf{e}^j), \tag{12}$$

Our goal is to select $Q$ dimensions out of $D$ to enhance the distinctiveness of the codes. We introduce a binary flag $\mathbf{F} \in \{0, 1\}^D$, where $F_k = 1$ ($k = 1, ..., D$) indicates that the $k^{th}$ dimension $\mathbf{e}_k^i$ is selected, and $\mathbf{F}\mathbf{F}^\top = Q$. Our objective now becomes finding the optimal $\mathbf{F}$ to minimize the Inter-Code Similarity:

$$\min_{\mathbf{F}} \quad S = \frac{1}{L^2} \sum_{i=1}^{L} \sum_{\substack{j=1 \\ j \neq i}}^{L} \delta(\mathbf{e}^i \odot \mathbf{F}, \mathbf{e}^j \odot \mathbf{F}), \tag{13}$$

where $\odot$ denotes element-wise multiplication.

We further suppose the Codebook has been L2-normalized, meaning that each code vector $\mathbf{e}^i \in \mathbb{R}^D$ has a unit length. Under this assumption, the cosine similarity between two code vectors $\mathbf{e}^i$ and $\mathbf{e}^j$ can be simplified as their dot product:

$$\delta(\mathbf{e}^i, \mathbf{e}^j) = \mathbf{e}^i \cdot \mathbf{e}^j, \tag{14}$$

where $\cdot$ denotes the dot product of two vectors. Then we can simplify the cosine similarity as

$$S = \sum_{k=d_1}^{d_Q} S_k = \sum_{k=d_1}^{d_Q} \left( \frac{1}{L^2} \sum_{i=1}^{L} \sum_{\substack{j=1 \\ j \neq i}}^{L} \mathbf{e}_k^i \cdot \mathbf{e}_k^j \right), \tag{15}$$

where $k = \{d_1, d_2, ..., d_Q\}$ denotes the indices of selected feature dimensions with $F_k = 1$, and $S_k = \frac{1}{L^2} \sum_{i=1}^{L} \sum_{\substack{j=1 \\ j \neq i}}^{L} \mathbf{e}_k^i \cdot \mathbf{e}_k^j$ represents the average inter-class similarity of the $k^{th}$ dimension. Through straightforward derivation, we observe that solving the optimization problem is equivalent to selecting Q elements with the smallest average similarity.

**Inter-code Variance:** Our objective is to minimize redundancy by eliminating feature dimensions with low variance across codewords, as such dimensions contribute limited discriminative information. In this framework, codewords are regarded as distinct semantic cluster centers. The variance for the $k^{th}$ feature dimension is formulated as:

$$V_k = \frac{1}{L} \sum_{i=1}^{L} (\mathbf{e}_k^i - \bar{\mathbf{e}}_k)^2, \tag{16}$$

where $\bar{\mathbf{e}}_k = \frac{1}{L} \sum_{i=1}^{L} \mathbf{e}_k^i$ represents the mean of the $k^{th}$ dimension across all codewords. Analogous to Inter-code Similarity, we select the $Q$ dimensions exhibiting the highest variance to augment discriminative power.

To amalgamate the criteria of similarity and variance, a balance factor $\lambda$ is introduced to compute the final metric for each feature dimension:

$$U_k = \lambda V_k - (1 - \lambda) S_k, \tag{17}$$

where $k = 1, \ldots, D$. The dimensions corresponding to the top-$Q$ biggest values of $U_k$ are chosen as the refined features, maximizing inter-class divergence and discrimination.

We conducted an evaluation of TOC on the open-source pre-trained model of DCID (Xia et al., 2024). The unified discrete representation space of this model is a Codebook consisting of 400 codewords, each with 256 dimensions. As depicted in Figure 2, the distinctiveness of the codes with the 128-dimensional features obtained after TOC computation is notably enhanced.

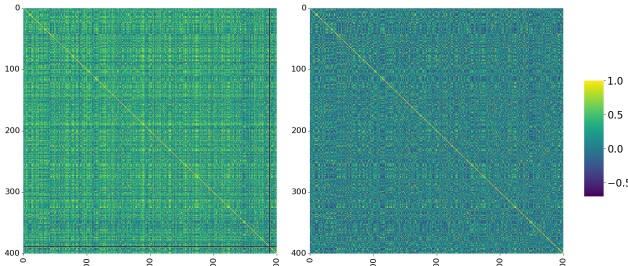

Figure 2: Left: Cosine similarity in the original codebook. Right: Cosine similarity after TOC.

# 4 EXPERIMENT

## 4.1 DATASETS AND TASKS

### 4.1.1 PRETRAIN

It consists of multimodal unified representation pre-training and Vector Quantized Variational AutoEncoder(VQVAE) (Van Den Oord et al., 2017) pre-training for single-modal images. **Multimodal Unified Representation:** The pretraining dataset uses the VGGsound-AVEL40K (Chen et al., 2020a; Zhou et al., 2022; 2021) with prompts provided by Xia et al. (2024). **Single-modal Representation:** We trained a VQVAE (Van Den Oord et al., 2017) on the CelebA-HQ 30K (Karras et al., 2017) dataset and tested the effects of using TOC to select certain feature dimensions for reconstruction. This evaluation assesses the ability of TOC to transfer to other domains involving the codebook.

### 4.1.2 DOWNSTREAM

The unified representation pre-trained models will be evaluated on several downstream tasks using different datasets. **Cross-modal event classification on AVE dataset:** (Tian et al., 2018) training on one modality (video) and evaluating on another (audio). **Cross-modal event localization on AVVP dataset:** (Tian et al., 2020) localizing events in one modality and transferring to the other. **Cross-dataset localization/classification:** training on classification in AVE and evaluating localization in AVVP, transferring across datasets. Cross-modal classification between UCF-101 (Soomro et al., 2012) visual clips and VGGSound-AVEL audio clips. **Cross-modal Zero-shot Retrieval:** we adopt a process similar to the test set (Yu et al., 2018) consists of 500 pairs from MSCOCO (Chen & Dolan, 2011), assesses the zero-shot retrieval capability for visual-text alignment. Clotho (Drossos et al., 2020); assesses the zero-shot retrieval capability for audio-text alignment. Flickr Sound (Senocak et al., 2018); assesses the zero-shot retrieval capability for audio-visual alignment. **Cross-modal Generation:** We modified the IP-Adapter (Ye et al., 2023) model by integrating our model as the image encoder and adding an MLP to align the dimensions with the IP-Adapter's input. We fine-tuned the model using 4,500 FlickrSound (Senocak et al., 2018) image-audio pairs over 80,000 steps with a batch size of 8, and tested it on other 500 pairs, evaluating both image to image, audio to image and text to image generation. For more details of downstream tasks, please refer to Appendix B.

## 4.2 IMPLEMENTATION DETAILS

The models we compare include the most outstanding recent developments in multimodal unified discrete representations and models that excel in multimodal domain generalization: CODIS (Duan et al., 2022), TURN (Zhao et al., 2022), CMCM (Liu et al., 2021a), SimMMDG (Dong et al., 2024), and DCID (Xia et al., 2024). These methods are implemented on our tasks, and their performance is evaluated on multi downstream tasks. For the AVE (Tian et al., 2018), VGGSound-AVEL (Zhou et al., 2022; 2021), and UCF101 (Soomro et al., 2012) datasets, precision is used as the metric. The F1-score is utilized for assessing the AVVP (Tian et al., 2020) and AVE→AVVP generalization task, and recall is utilized for zero-shot retrieval (Chen & Dolan, 2011; Drossos et al., 2020). Mean Square Error (MSE) is employed to evaluate the reconstruction quality of TOC on the CelebA-HQ

30K dataset (Karras et al., 2017). Additionally, Fréchet Inception Distance (FID) (Heusel et al., 2017) is used to assess the model's capability in cross-modal generalization.

In the TOC formulation, the parameter $\lambda$ is set to 0.3, and in $L_{nce}$, the parameter $\tau$ is set to 1.0. All results presented in table 1, 2, 4, 5, 6 were obtained with a codebook size set to 400 and an embedding dimension set to 256. The table 3 involves VQVAE with a codebook size of 128 and an embedding dimension of 128. The ablation study on codebook size is discussed in Table 7. The backbone models used to extract features for video, audio, and text modalities are VGG19 (Simonyan & Zisserman, 2014), VGGish (Hershey et al., 2017), and BERT (Devlin et al., 2018), respectively.

### 4.3 Performance Analysis

In the tables below, **bold** numbers indicate the best results, while green values in parentheses show the performance improvement attributed to the TOC.

Table 1: Comparison with SOTA methods on four audiovisual downstream tasks. (SimMMDG represents recent great work in multimodal domain generalization; however, it does not utilize discrete representations, making it incompatible with TOC for optimization.)

| Method | AVE | | AVVP | | AVE→AVVP | | UCF(v)↔VGG(a) | | Avg. |
|---|---|---|---|---|---|---|---|---|---|
| | V→A | A→V | V→A | A→V | V→A | A→V | V→A | A→V | |
| CODIS (Duan et al., 2022) | 36.8 | 39.7 | 32.7 | 32.6 | 40.8 | 40.6 | 50.8 | 45.2 | 39.90 |
| TURN (Zhao et al., 2022) | 37.6 | 39.2 | 32.4 | 32.2 | 40.6 | 41.4 | 50.4 | 46.1 | 39.99 |
| CMCM (Liu et al., 2021a) | 46.3 | 45.8 | 36.1 | 35.2 | 47.1 | 48.2 | 51.2 | 48.3 | 44.78 |
| SimMMDG (Dong et al., 2024) | 49.5 | 51.7 | 39.3 | 39.7 | 52.9 | 52.7 | 64.5 | 58.8 | 51.14 |
| DCID (Xia et al., 2024) | 54.1 | **55.0** | 40.4 | 40.8 | 53.0 | 52.4 | 67.1 | 60.6 | 52.93 |
| FCCID | **55.2** | 54.9 | **42.4** | **44.5** | 55.3 | 57.4 | **69.4** | 61.6 | **55.09** |
| CODIS+TOC | 37.2 | 41.3 | 33.1 | 33.9 | 41.9 | 42.4 | 51.2 | 47.3 | 41.04(+1.14) |
| TURN+TOC | 38.3 | 40.5 | 33.2 | 32.9 | 41.5 | 43.3 | 51.5 | 46.8 | 41.00(+1.01) |
| CMCM+TOC | 46.9 | 47.2 | 37.9 | 36.2 | 49.8 | 50.1 | 52.3 | 49.1 | 46.19(+1.41) |
| DCID+TOC | 54.5 | **55.0** | 40.9 | 41.6 | 56.5 | 53.6 | 68.1 | 61.7 | 53.99(+1.06) |
| FCCID+TOC | **55.9** | **55.0** | **43.6** | **45.1** | **57.4** | **58.5** | **69.6** | **62.0** | **55.89**(+0.80) |

Table 2: Comparison with SOTA methods on three cross-modal zero-shot retrieval tasks, all results are calculated as the mean across two directions.

| Method | MSCOCO(V↔T) | | | Clotho(A↔T) | | | FlickrSound(V↔A) | | | Avg. |
|---|---|---|---|---|---|---|---|---|---|---|
| | R@1 | R@5 | R@10 | R@1 | R@5 | R@10 | R@1 | R@5 | R@10 | |
| CMCM (Liu et al., 2021a) | 0.50 | 4.20 | 7.20 | 1.62 | 8.04 | 14.87 | 2.20 | 9.80 | 15.60 | 7.11 |
| DCID (Xia et al., 2024) | 0.80 | **5.00** | 8.30 | 2.06 | 9.00 | 16.70 | **3.10** | 11.10 | 17.20 | 8.14 |
| FCCID | **1.30** | 4.90 | **9.60** | **2.87** | 10.73 | **18.19** | **3.10** | 11.80 | 17.50 | **8.89** |
| CMCM+TOC | 0.70 | 4.50 | 7.70 | 1.93 | 8.43 | 15.33 | 2.40 | 10.60 | 16.10 | 7.52(+0.41) |
| DCID+TOC | 1.10 | **5.30** | 8.80 | 2.59 | 9.00 | 17.08 | 3.60 | 11.80 | 17.80 | 8.56(+0.42) |
| FCCID+TOC | **1.50** | 5.10 | **10.40** | **3.16** | **11.15** | **19.04** | **3.80** | **12.20** | **18.40** | **9.42**(+0.53) |

**TOC:** As shown in table 1 and table 2, TOC optimizes methods with discrete representation spaces, facilitating at least a 0.80% improvement in average results for cross-modal generalization tasks, and a minimum average increase of 0.41% for cross-modal zero-shot retrieval tasks. Notably, these results are achieved with just a single refinement of the codebook, requiring no more than 10 seconds. Moreover, due to the reduced dimensions of the refined codebook, fewer parameters need to be trained for downstream tasks, which accelerates the evaluation speed. As shown in table 4 and figure 4, it also excels in cross-modal generation tasks, aiding the model in enhancing both image-to-image ($I \rightarrow I$), audio-to-image ($A \rightarrow I$) and text-to-image ($T \rightarrow I$) generation outcomes.

We have also explored extending the TOC to unimodal discrete representation space. Using a VQ-VAE model trained on the CelebA-HQ 30K dataset (Karras et al., 2017), we tested reconstruction results using only a subset of the codeword's dimensions. As shown in Table 3, R100-avg represents the average outcome of 100 random selections of codeword dimensions for reconstruction, where TOC masks the least important dimensions. 'Count' indicates the number of times out of these 100 trials that the MSE was greater than the MSE for TOC. The MSE for TOC reconstructions with only 25.0% to 87.5% of the dimensions was significantly lower than the average MSE of 100 random selections. Moreover, the last column indicates that TOC is statistically superior to random selection. Partial demonstration results are shown in Figure 3, For all columns except "origin", the left

half of each image shows reconstructions with randomly masked codeword dimensions, while the right half shows reconstructions after masking the least important codeword dimensions with TOC, it is evident that the dimensions selected by TOC are significantly more effective than those chosen randomly. For more results for image reconstruction, please refer to Appendix F.1.

Table 3: Comparison of image reconstructions using random masking versus TOC masking.

| Mask (%) | R100-avg↓ | TOC↓ | Count↑ |
|---|---|---|---|
| 87.5 | 0.0621 | 0.0231 | 100 |
| 75.0 | 0.0477 | 0.0159 | 100 |
| 62.5 | 0.0335 | 0.0109 | 100 |
| 50.0 | 0.0229 | 0.0086 | 100 |
| 37.5 | 0.0141 | 0.0062 | 96 |
| 25.0 | 0.0075 | 0.0039 | 90 |

Table 4: Performance on cross-modal generalization

| Method | I2I↓ | A2I↓ | T2I↓ |
|---|---|---|---|
| CMCM | 129.56 | 130.93 | 148.93 |
| DCID | 121.44 | 123.28 | 141.16 |
| FCCID | **116.06** | **117.26** | **135.52** |
| CMCM+TOC | 124.25 | 125.37 | 144.93 |
| DCID+TOC | 118.30 | 119.96 | 135.93 |
| FCCID+TOC | **113.95** | **115.14** | **130.98** |

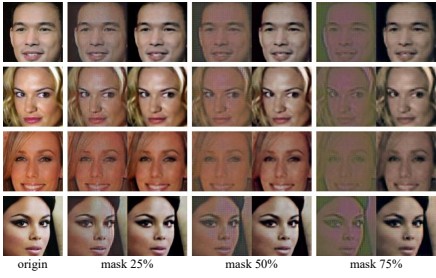

origin      mask 25%      mask 50%      mask 75%

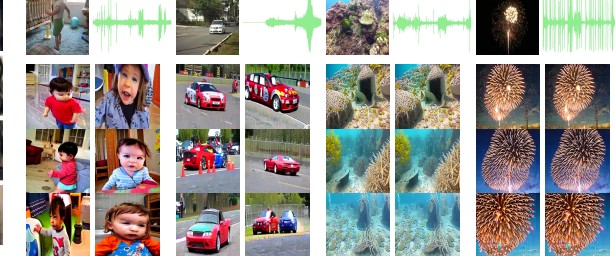

Figure 3: Example results of reconstructions using random and TOC masking.

Figure 4: Example results of cross-modal image generation experiments conducted by FCCID+TOC.

**FCCID:** Reviewing Tables 1 and 2 clearly shows that FCCID and FCCID+TOC consistently outperform all other methods across a variety of tasks. Compared to the previous SOTA, FCCID achieves an average improvement of 2.16% in four cross-modal generalization tasks and an average improvement of 0.75% in three cross-modal zero-shot retrieval tasks. As shown in Table 4, these approaches also demonstrate a clear advantage in cross-modal generation tasks. All results suggest that our methods can more effectively process and understand cross-modal information.

As illustrated in Figure 4, the top row displays four pairs of image-audio samples, while the three rows below show images generated based on these samples. It is observable that FCCID, even when trained only with images, can achieve A → I results, closely resembling the I → I outcomes, especially in the last two examples where the generated images are identical. This indicates that these two pairs of image-audio samples are mapped to the same code in the codebook, demonstrating a high degree of modal alignment. For additional examples and results for T → I, please refer to Appendix F.2.

We further demonstrate multimodal quantization activations for the discrete representation spaces of both DCID and FCCID. FCCID shows significantly better quantization consistency across different modalities compared to DCID. Detailed results can be found in Appendix E.

**Ablation Study:** The two critical modules in FCCID are the disentanglement components of FCID and CCID. Given that alignment is crucial for unified representations, it is unnecessary to conduct ablation studies on this aspect. Other loss functions derived from prior work will not be discussed here. Therefore, the ablation studies on FCCID will focus exclusively on the most important disentanglement components, namely $\hat{I}_{vCLUB_f}$ and $\hat{I}_{vCLUB_c}$, which are composed of $A_{CLUB}, V_{CLUB}$ and $AV_{CLUB}, TE_{CLUB}$, respectively. Both components of TOC are novel contributions by us, and we have conducted ablation studies on them within the FCCID model.

Table 5 demonstrates that $A_{CLUB}$ and $V_{CLUB}$ have a more significant impact on the model's performance in AV-related downstream tasks, which is evident. Additionally, $TE_{CLUB}$ also affects the

model results to some extent, as the textual information may contain irrelevant and missing AV data that can influence the outcomes if not properly disentangled. Similarly, when using only $AV_{CLUB}$, the audiovisual features extracted by the model still contain a degree of audio-specific and video-specific information. The disentanglement provided by $AV_{CLUB}$, along with the alignment between audiovisual and text, helps to separate this information to some extent.

As shown in Table 6, the two components of TOC individually contributed to an average improvement of 0.54% and 0.10% across eight metrics for FCCID, and when combined, further enhanced the average results by 0.80%. This demonstrates that both components of TOC are effective and, when used together, yield better performance.

Table 7 presents the performance of the FCCID model across various codebook sizes. It is observed that the model achieves the best average results when the codebook size is set to 400. Conversely, using either a excessively large or small codebook size may lead to insufficient semantic learning or inadequate semantic expression, resulting in decreased model performance.

Table 5: Ablation studies on the impact of FCCID

| $A_{CLUB}$ | $V_{CLUB}$ | $AV_{CLUB}$ | $TE_{CLUB}$ | AVE | | AVVP | | AVE→AVVP | | UCF(v)↔VGG(a) | | Avg. |
|---|---|---|---|---|---|---|---|---|---|---|---|---|
| | | | | V→A | A→V | V→A | A→V | V→A | A→V | V→A | A→V | |
| - | - | - | - | 51.3 | 51.6 | 39.5 | 40.7 | 50.6 | 51.1 | 63.3 | 57.6 | 50.71 |
| ✓ | - | - | - | 52.4 | 53.5 | 40.9 | 42.4 | 53.1 | 54.2 | 66.0 | 59.8 | 52.79 |
| - | ✓ | - | - | 53.1 | 53.4 | 41.7 | 43.2 | 53.9 | 54.7 | 67.1 | 60.1 | 53.40 |
| - | - | ✓ | - | 52.2 | 51.9 | 40.2 | 41.7 | 52.4 | 52.5 | 64.2 | 59.1 | 51.78 |
| - | - | - | ✓ | 51.7 | 51.5 | 40.6 | 41.8 | 52.5 | 52.9 | 63.5 | 58.2 | 51.59 |
| ✓ | ✓ | - | - | 54.2 | 54.0 | 41.4 | 43.9 | **55.9** | 56.1 | 67.9 | 61.3 | 54.34 |
| - | - | ✓ | ✓ | 52.9 | 52.6 | 40.8 | 42.1 | 52.5 | 53.9 | 65.7 | 59.2 | 52.46 |
| ✓ | ✓ | ✓ | ✓ | **55.2** | **54.9** | **42.4** | **44.5** | 55.3 | **57.4** | **69.4** | **61.6** | **55.09** |

Table 6: Ablation studies on the impact of TOC

| Inter-code Similarity | Inter-code Varience | AVE | | AVVP | | AVE→AVVP | | UCF(v)↔VGG(a) | | Avg. |
|---|---|---|---|---|---|---|---|---|---|---|
| | | V→A | A→V | V→A | A→V | V→A | A→V | V→A | A→V | |
| - | - | 55.2 | 54.9 | 42.4 | 44.5 | 55.3 | 57.4 | 69.4 | 61.6 | 55.09 |
| ✓ | - | 55.8 | 54.5 | **43.6** | **45.7** | 56.8 | 58.3 | 69.2 | 61.1 | 55.63 |
| - | ✓ | 55.6 | **55.0** | 43.4 | 44.8 | 56.2 | 54.8 | **69.8** | 61.9 | 55.19 |
| ✓ | ✓ | **55.9** | **55.0** | **43.6** | 45.1 | **57.4** | **58.5** | 69.6 | **62.0** | **55.89** |

Table 7: Ablation studies on the impact of Codebook Size

| Codebook Size | AVE | | AVVP | | AVE→AVVP | | UCF(v)↔VGG(a) | | Avg. |
|---|---|---|---|---|---|---|---|---|---|
| | V→A | A→V | V→A | A→V | V→A | A→V | V→A | A→V | |
| 256 | 52.9 | 52.3 | 38.8 | 43.2 | 53.7 | 53.9 | **70.8** | 56.4 | 52.75 |
| 300 | 52.8 | 54.1 | 42.1 | 44.1 | 54.1 | **58.5** | 69.6 | 60.4 | 54.46 |
| 400 | **55.2** | **54.9** | **42.4** | **44.5** | **55.3** | 57.4 | 69.4 | **61.6** | **55.09** |
| 512 | 54.4 | 52.4 | 40.0 | 42.6 | 54.1 | 56.9 | 70.3 | 59.3 | 53.75 |
| 800 | 52.2 | 54.6 | 41.6 | 43.9 | 53.1 | 56.7 | 69.6 | 59.7 | 53.93 |
| 1024 | 52.8 | 54.5 | 40.4 | 41.6 | **55.3** | 55.9 | 65.8 | 58.6 | 53.11 |

## 5 CONCLUSION

Inspired by works on feature importance and training-free optimization, we propose TOC. This is the first application of training-free optimization to the discrete representation space, enhancing multimodal and single-modal (e.g., images) representations. We also introduce the FCCID framework. Unlike previous research in the domain of unified discrete representations that often overlooked modal differences, our method starts from the temporal characteristics of audiovisual data and the distinct nature of text. We significantly enhance the effectiveness of unified representations through two different granularities of disentanglement.

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

## A  RELATED WORK

**Multi-Modal Unified Representation:** In recent years, significant efforts have been directed towards developing multi-modal unified representations. This includes approaches that implicitly align different modalities into a shared latent space (Petridis et al., 2018; Sarkar & Etemad, 2022; Andonian et al., 2022) and strategies that train modal-general encoders to extract information across modalities (Chen et al., 2020b; Wang et al., 2022). Techniques such as cross-modal knowledge distillation facilitate knowledge transfer between modalities (Sarkar & Etemad, 2022; Pedersoli et al., 2022). Additionally, several works have connected continuous representation spaces of various modalities through bridging techniques, thereby leveraging the strengths of different models to achieve superior unified representations across multiple modalities (Wang et al., 2023b;a; Wang et al.). At the same time, to enhance interpretability, unified expressions are often constructed using codebooks or prototypes (Duan et al., 2022; Lu et al., 2022; Liu et al., 2021a; Zhao et al., 2022; Xia et al., 2024). For instance, Duan et al. (2022) employs Optimal Transport to map feature vectors from different modalities to prototypes, while Zhao et al. (2022) utilize self-cross-reconstruction to enhance mutual information. Liu et al. (2021a) implement a similar scheme to align videos with speech and text, although they assume perfect alignment between modalities. Addressing the challenge of non-perfectly aligned multimodal sequences, Xia et al. (2024) map these sequences into a common discrete semantic space through rational information decoupling. To address the lack of attention to the inherent differences among modalities in previous works, we introduce the FCCID framework, which performs alignment and information disentanglement at varying granularities while considering the distinctions between text and audiovisual modalities.

**Training Free Optimization:** Recent works have explored diverse approaches to enhance model performance without additional training. The Training-Free CLIP-Adapter (Tip-Adapter) (Zhang et al., 2022) and the Adaptive Prior rEfinement (APE) method (Zhu et al., 2023) leverage non-parametric and refinement techniques, respectively, to improve few-shot classification capabilities of the CLIP model. In diffusion models, a novel training-free method (Chen et al., 2024a) optimizes time steps and model architecture for efficient image generation, while the FuseDream pipeline (Liu et al., 2021b) employs a CLIP+GAN approach for robust text-to-image generation. Beyond CLIP-based models, the TEEN method (Wang et al., 2024) offers a training-free approach for few-shot class-incremental learning, efficiently recognizing new classes without training costs. Recently, there has been a surge of research in popular areas such as video generation (Chen et al., 2024b; Yang et al., 2024; Peng et al., 2024; Zhang et al., 2023) and multimodal large language models (Wu et al., 2024), where several works have attempted to enhance model performance through training-free methods. Building on these advancements, this paper introduces TOC, which, to the best of our knowledge, represents the first exploration of training-free optimization within the context of multimodal unified discrete representation. This work further extends the scope of training-free approaches in the field.

## B  BACKGROUND

**Cross Modal Generalization (CMG)** is a task introduced by Xia et al. (2024) that evaluates the model's ability to map diverse modalities, such as text, audio, and video, into a unified discrete latent space. The model's ability for cross-modal zero-shot knowledge transfer is evaluated through a setup where training is conducted on modality $m1$ and testing is performed on modality $m2$.

During training, the model learns a representation for inputs from one modality using the encoder $\Phi^{m1}$ and the downstream decoder $\mathbf{D}$:

$$\mathbf{E}(\mathbf{D}(VQ(\Phi^{m1}(\mathbf{x}_i^{m1}))), \mathbf{y}_i^{m1}), \tag{18}$$

where $\mathbf{x}_i^{m1}$ is the input, $\mathbf{y}_i^{m1}$ is the label, and $\mathbf{E}$ is the evaluation function. During testing, the model is evaluated on a different modality $m2$, demonstrating its ability to generalize:

$$\mathbf{E}(\mathbf{D}(VQ(\Phi^{m2}(\mathbf{x}_i^{m2}))), \mathbf{y}_i^{m2}). \tag{19}$$

Here, $m1, m2 \in a, b, c$ and $m1 \neq m2$. The parameters of both $\Phi^{m1}$ and $\Phi^{m2}$ are parameters frozen during training and testing, while only the parameters of $\mathbf{D}$ are updated during training.

**Dual Cross-modal Information Disentanglement(DCID)** (Xia et al., 2024) is a framework designed to align primary common events across modalities by disentangling and refining shared se-

mantic content within cross-modal data. It employs modal-specific encoders $\Psi^m$ to extract modal-specific features $\overline{\mathbf{z}}_i^m$ and modal-general encoders $\Phi^m$ to extract modal-general features $\mathbf{z}_i^m$ from modalities $m \in \{a, b, c\}$. The framework optimizes mutual information between these features to minimize redundancy and enhance semantic alignment.

**Mutual Information Minimization:** DCID utilizes the CLUB (Cheng et al., 2020) method to minimize the mutual information between modal-general and modal-specific information within each modality:

$$
\hat{I}_{\text{vCLUB}} = \frac{1}{N} \sum_{i=1}^{N} \left[ \log q_\theta(\overline{\mathbf{z}}_i^m | \mathbf{z}_i^m) - \frac{1}{N} \sum_{j=1}^{N} \log q_\theta(\overline{\mathbf{z}}_j^m | \mathbf{z}_i^m) \right], \tag{20}
$$

where $q_\theta$ is the variational approximation of the ground-truth posterior, $N$ is the number of samples, and $m$ denotes the modality.

**Mutual Information Maximization:** To maximize mutual information across different modalities, DCID employs Cross-Modal CPC (Oord et al., 2018), predicting future samples in one modality using context representations from another modality. The objective is formulated as:

$$
L_{\text{cpc}} = -\frac{1}{R} \sum_{r=1}^{R} \log \left[ \frac{\exp(\mathbf{z}_{t+r}^n W_r^m \mathbf{o}_t^m)}{\sum_{\mathbf{z}_j \in Z_{neg}} \exp(\mathbf{z}_j^n W_r^m \mathbf{o}_t^m)} \right], \tag{21}
$$

where $W_r^m$ is a learnable weight matrix, $\mathbf{o}_t^m$ is the context representation, $R$ is the prediction horizon, $t$ is the time step, and $Z_n$ is a set of negative samples.

## C LIMITATIONS

FCCID is specifically designed for scenarios involving tri-modal representations encompassing audio, video, and text. In contrast, when operating with only two modalities, the model can utilize either FCID or CCID. The theoretical framework underlying TOC is based on several assumptions that hold under ideal conditions, which highlights a potential area for future enhancement. This suggests that while TOC effectively addresses certain challenges in multi-modal alignment, there remains room for refinement and further development to improve its robustness in more varied real-world conditions.

## D COMPUTER RESOURCES

Training the complete FCCID model using a single Nvidia RTX 3090 GPU takes 10 hours, while TOC requires no additional training. Training the VQVAE model in this paper takes 1 hour on a single Nvidia RTX 3090, and 24 hours if PixelCNN is included. All individual downstream experiments can be completed within 1 hour. The parameter count for the FCCID Encoder (including the codebook) is 78M, while the DCID (Xia et al., 2024) Encoder (including the codebook) has 80M parameters. Compared to previous SOTA models, we achieve superior unified representation performance with a reduced parameter count.

## E ACTIVATION OF CODEBOOK

As shown in Figures 5 and 6, we utilize the audio-video-text tri-modal data from VALOR32K Chen et al. (2023) to quantify the codes in the DCID and FCCID codebooks. In these figures, red points indicate that the activation frequency of a single modality $> 95\%$, green points denote that the activation counts for all three modalities are $\geq 5\%$, while blue points fall between the two categories. The images clearly demonstrate that FCCID exhibits significantly better alignment across the three modalities compared to DCID, with a notable reduction in codes activated solely by a single modality. This highlights the enhancement FCCID provides for unified representations in the tri-modal context.

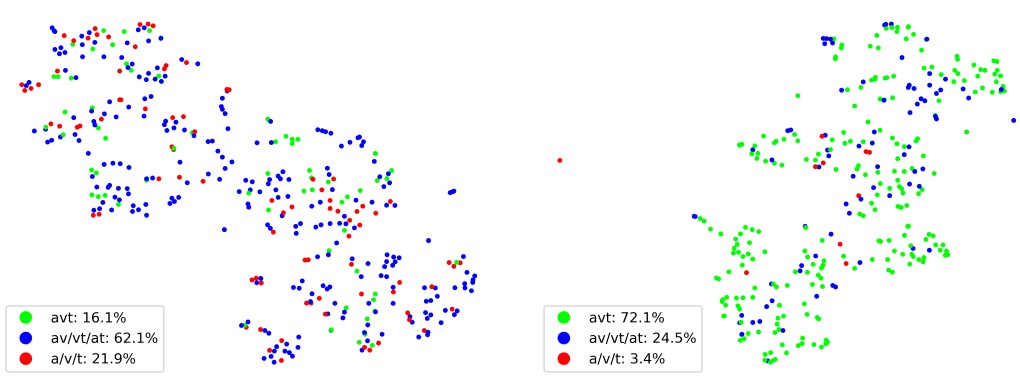

Figure 5: DCID's codebook activate        Figure 6: FCCID's codebook activate

## F MORE RESULT ABOUT RECONSTRUCTION AND GENERATION

### F.1 RECONSTRUCTION

As shown in Figure 7, for all columns except the 'origin' column, the images on the left represent reconstructions with random masks, while the images on the right illustrate reconstructions using the dimensions with the highest TOC retention scores. It is evident that TOC significantly outperforms random masking in reconstructions with mask ratios ranging from 25.0% to 87.5%, with the performance gap becoming increasingly pronounced as the mask ratio increases.

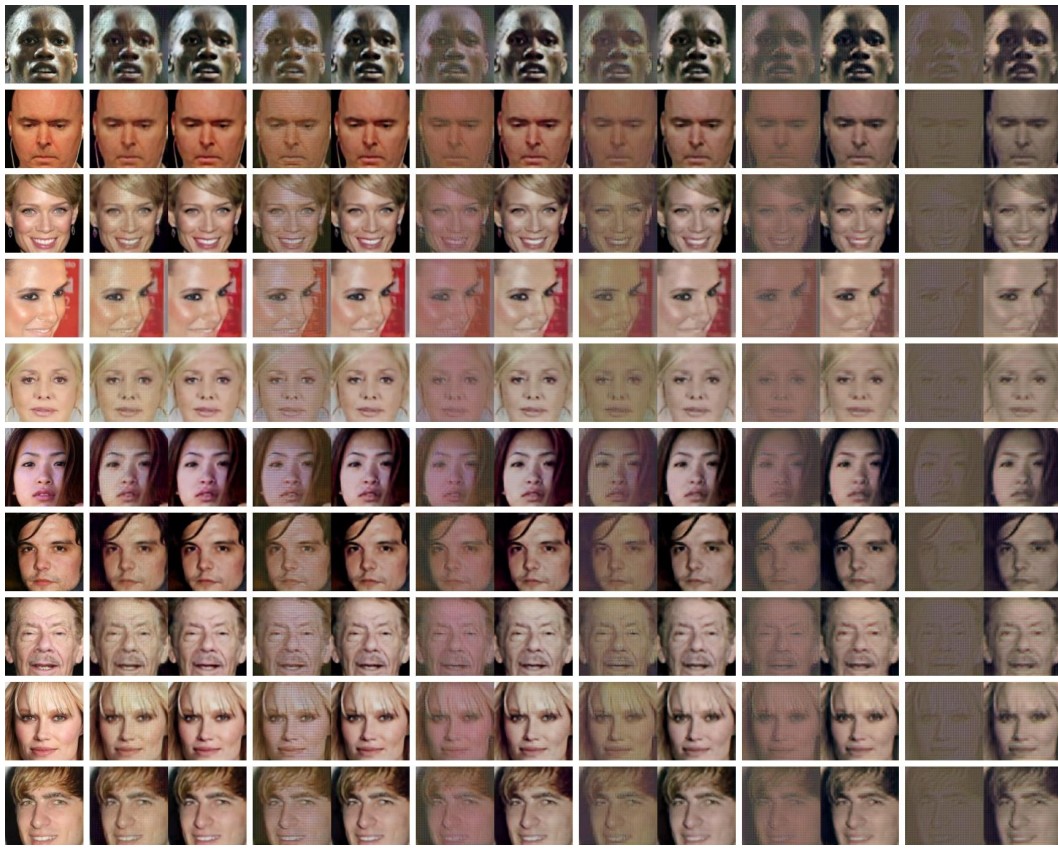

origin     mask 25.0%     mask 37.5%     mask 50.0%     mask 62.5%     mask 75.0%     mask 87.5%

Figure 7: More results of reconstructions using random and TOC masking.

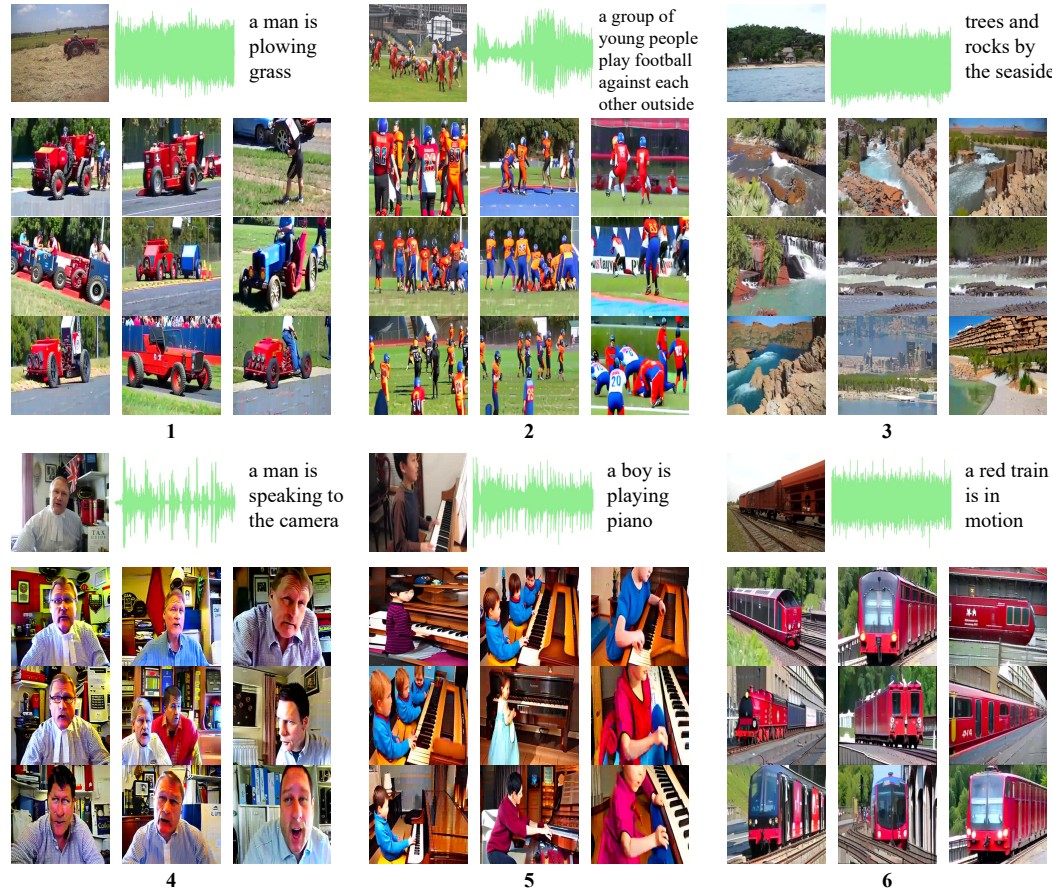

Figure 8: More results of Cross-modal generation.

## F.2 GENERATION

As shown in Figure 8, thanks to multimodal unified representations, the results of cross-modal image generation from audio and text closely resemble actual images. As evident in samples 2 and 6, despite the audio not mentioning specific details such as the color of clothing and trains, these elements are still accurately generated, which can be attributed to the discrete unified representation serving as a central semantic hub for multiple modalities. In contrast, the results from Text-to-Image (T→I) are noticeably inferior to those from Image-to-Image (I→I) and Audio-to-Image (A→I). This difference is exemplified in the first image generated from sample 1's text, where the action of a car mowing grass is mistakenly transformed into a man mowing grass. This discrepancy arises because the semantic connections between images and audio are stronger than those generated through model-based text, which merely mentioned 'man' and 'plowing grass' without specifying the tool used for plowing.

