# OpenReview forum: "Advancing Multimodal Unified Discrete Representations"
_ICLR.cc/2025/Conference — ICLR 2025 Conference Withdrawn Submission_

### Official Review · Reviewer_8kwo · 2024-10-27

**Soundness:** 2
**Presentation:** 3
**Contribution:** 2
**Rating:** 5
**Confidence:** 3

**Summary:**

This paper proposes a general multimodal alignment method that can align audio, video and text modalities. As described in the paper, there are two main motivations. The first motivation is that current cross-modality alignment methods often lose the unique information of each modality during the alignment process. Therefore, the proposed method uses a separate encoder for the two modalities of video and audio to extract modality-specific features, respectively, in addition to the model-general encoders for the two modalities to perform alignment. The second motivation is that current alignment methods directly calculate the Euclidean distance between the embeddings extracted by encoders, which actually assumes that every dimension of the extracted embeddings contains useful information. The embeddings are often sparse only the information and there exist redundant dimensions. This paper proposes TOC to identify the importance of feature dimensions without additional training. The proposed method is pre-trained on the VGGSound dataset, and then the pre-trained model is applied to different downstream tasks such as cross-modal zero-shot retrieval and cross-modal generation.

**Strengths:**

1. The motivation is clear and reasonable.
2. The proposed training-free optimization of the codebook (TOC)  can assess the importance of feature dimensions without the need of additional training, which is technically sound.
3. Experimental evaluations are conducted on different downstream tasks such as cross-modal retrieval and modality generation.

**Weaknesses:**

1. Novelty: After extracting features with existing models, the proposed method combines the existing CLUB and CPC frameworks and VQVAE to construct its architecture. The idea of selecting informative dimensions seems to follow APE (Zhu et al., 2023). The proposed TOC follows the similar idea of using inter-class distances and intra-class variances to reduce dimensions as in LDA (linear discriminant analysis). Therefore, the novelty of the proposed method seems to be limited.
2. Experimental evaluations: The proposed method is evaluated on audio and video retrieval tasks as well as conditional image generation tasks on specific datasets. However, there are only a few comparison methods. Especially for generation tasks, the proposed method is mainly compared with VQ-VAE. At present, most state-of-the-art methods are based on diffusion models. Therefore, the experimental evaluation is insufficient, and experimental comparisons with more state-of-the-art methods are needed to demonstrate the effectiveness of the proposed method.
3. In terms of ablation studies, the authors verified the effects of different modules. However, most of these modules already exist.
4. The paper tests different quantities of embeddings in the codebook, but since the focus is on leveraging certain effective features from the embeddings, it would be better to include corresponding evaluations on different embedding dimensions.

**Questions:**

Although the motivation is reasonable, the novelty of the proposed method is somewhat incremental and the experimental evaluations are not fully convincing since some important aspects are lacking. Please refer to the weaknesses.

---

### Official Review · Reviewer_qU3n · 2024-10-29

**Soundness:** 3
**Presentation:** 2
**Contribution:** 2
**Rating:** 5
**Confidence:** 3

**Summary:**

The paper introduces two modules, FCCID and TOC, to address the issue that a unified alignment method cannot capture modality-specific features and that quantizing with Euclidean distance vectors neglects the distinctions between dimensions, resulting in redundant representations. The authors conducted sufficient experiments and ablation studies to demonstrate the effectiveness of their proposed method.

**Strengths:**

1. The authors conduct plenty experiments to support their method, and the results are significant.
2. The TOC module is novel, which incorperates the importance of feature dimensions into the calculation of the codebook.

**Weaknesses:**

1. I believe that the content of the related work section should ideally be included in the main text rather than the appendix, even though I understand there are page limitations.
2. In Equation 9, $sg$, $\phi_k^m$ are not explained.
3. The calculation of $U_k$ in Equation (9) and how it is updated require further clarification.
4. Some obvious formatting errors need to be corrected, such as the missing letter 'n' in "Sectio3.1" on line 125 and the missing closing parenthesis in the expression $p(x,y$ in Equation 2. Besides, in line 428, the ‘Count’ should be `Count’.
5. How to obtain $o_m^t$ needs to be clarified in line 174.
6. The article's innovation does not seem very strong. Compared to the method by Xia in the 2023 NIPS, it only adds a text modality and a coarse-grained contrastive learning approach. It seems that only the proposed TOC is novel.

**Questions:**

See the weaknesses.

---

### Official Review · Reviewer_sHST · 2024-11-01

**Soundness:** 2
**Presentation:** 1
**Contribution:** 2
**Rating:** 3
**Confidence:** 4

**Summary:**

The paper introduces FCCID, a model that achieves fine-grained alignment and disentanglement of audiovisual data, followed by compression. Experiments conducted across various cross-modal generalization setups demonstrate performance improvements of 1-2% over previous best-performing methods.

**Strengths:**

The proposed method addresses a crucial challenge of preserving temporal information in audiovisual data while disentangling unique characteristics of text and audiovisual modalities. It also explores a training-free approach to optimizing the codebook. These contributions are likely to be of interest to the ICLR community.

**Weaknesses:**

* **Clarity and Self-Containment**: Many sections require additional elaboration to make the paper self-contained and accessible to readers:
   - **Equation 2**: The distinction between the two representations beyond using different encoders is unclear. More detail on the objectives and the roles of each representation is needed. For instance, why one representation is model-general and the other model-specific, despite both using the same input data. Further clarification is also needed on whether the specific encoders are pre-trained or jointly trained with the rest of the model.
   - **Equation 3**: The transformation from $q_{\theta}(y|x)$ to $q_\theta(\bar{f}_i^m |{f}_i^m)$, along with the second part of the transformation, is hard to parse. The same question persists for Equation 6.
   - **Equation 4**: It is unclear how Equations 3 and 4 integrate into the final learning objective and why Equation 4 relies solely on model-general features rather than including model-specific ones as well. Equation 7 raises similar questions.
   - **Combination of Losses in Equation 10**: Additional detail on how the losses are combined, the purpose of each, and whether hyper-parameters are used or if they are equally weighted would enhance understanding.
   - (Minor) There is a missing parenthesis in Equation 2 in $p(x,y)$.

* **Experimental Design**:
   - **Uni-Modal Baselines**: For non-retrieval methods, including uni-modal baselines would help readers assess the standalone performance of methods that focus exclusively on one modality.
   - **Confidence Intervals**: Confidence intervals for the results would help quantify the benefits of TOC, which are currently difficult to justify without them, especially in Tables 1, 2, and 6.
   - **Codebook Size**: The authors mention that increasing the codebook size beyond 400 degrades performance, but more discussion is needed to clarify why this occurs.

**Questions:**

Please refer to my comments in the Weaknesses section.

---

### Official Review · Reviewer_cb3g · 2024-11-02

**Soundness:** 3
**Presentation:** 2
**Contribution:** 2
**Rating:** 5
**Confidence:** 4

**Summary:**

This paper proposes two methods for learning multi-modal unified representations. The Fine and Coarse Cross modal Information Dissentancing (FCCID) method imitates DCID to constrain and align the features output by semantic encoders and modality specific encoders of different modalities, extending the work of DCID on two modalities to three modalities. The Training-free Optimization of Codebook (TOC) method selects codebook’s dimensions with the goal of simultaneously minimizing similarity and maximizing variance to remove redundancy. Through these two methods, the article achieved the learning and de redundancy of multi-modal unified representation. The article has demonstrated the superiority of this method through experiments.

**Strengths:**

1.The problem that the article aims to solve is meaningful

2.The experimental results of the article are good

3.The experiment was conducted sufficiently

**Weaknesses:**

1.In terms of writing, the symbols in Figure 1 cannot be displayed on some PDF readers

2.Lack of innovation, the similarity between the two methods and previous work like DCID and APE is too great

3.Lack of an algorithm description to assist in understanding the entire training process

**Questions:**

1.What are the specific loss items in the CLUB between video and audio, text and audio-visual modalities in the ablation experiment? It was not mentioned earlier.

2.How to fine-tune downstream tasks with less than three modalities?

---

### Official Review · Reviewer_LfEr · 2024-11-03

**Soundness:** 3
**Presentation:** 2
**Contribution:** 3
**Rating:** 5
**Confidence:** 4

**Summary:**

The authors identify that  previous works researching on multi-modal unified representation may overlook the different granularity of different modals when aligning them, also the redundant background info in the shared information across modalities in discrete multi-modal unified representation, and the overlook of disentanglement of different feature dimensions.
To alleviate them, the authors propose FCCID and TOC for disentanglements targeting on different purposes.

**Strengths:**

The authors propose two modules for disentangling learning for different purposes as illustrated, which are novel.

The modules proposed can enhance the performance of accuracy and efficiency, as also analyzed and demonstrated.

**Weaknesses:**

The writing should be improved. For example, “Specifically, FCCID, TOC, and their combination outperformed before SOTA by 2.16%, 1.06%, and 2.96% respectively, on four downstream tasks.” Whether these percentages represent average improvements across the four tasks?  Besides, some writing errors need to be corrected, like line 39: has-> have, in line 125, Sectio -> Sectio, etc.

In line 83, could the authors point out what the relationship between “the inherent constraints of the codebook and the quantization method based on Euclidean distance.” and “treats all feature dimensions equally”, it’s rather unclear about the relation here.

Is lamda a hyper-parameter? How is the lamda value chosen? It’s better for the authors to include a hyper-parameter investigation in their manuscripts.

“TOC reduces training parameters and time”: It could be better for the authors to include a table or figure comparing the number of parameters and training time with and without TOC across different models or tasks.

It’s better for the authors to re-organize the layout of some figures and tables and introduce their related works in the main text for better review. Like using some techniques such as wrap figure in latex.

**Questions:**

See weaknesses.

---

### Official Review · Reviewer_mtj9 · 2024-11-03

**Soundness:** 2
**Presentation:** 3
**Contribution:** 2
**Rating:** 3
**Confidence:** 5

**Summary:**

This paper mainly focuses on the learning of multimodal discrete unified representations. To address this problem, this paper proposed two methods FCCID and TOC, aiming to perform fine and coarse disentanglement of information and refine the unified discrete representations obtained from pre-training. Experimental results show the effectiveness of the proposed methods.

**Strengths:**

The experimental results show the effectiveness of the proposed methods. Also, the proposed training-free TOC is interesting, TOC focuses on optimizing the pre-trained codebook and performs calculations independently of downstream information.

**Weaknesses:**

1. The contribution of this paper is very limited. A very serious problem is that this article is very similar to article [1] published in NeurIPS 2024, and the similarities include language expressions, symbols, formulas, and so on.

[1] Achieving Cross Modal Generalization with Multimodal Unified Representation, NeurIPS 2024.

2. Considering the first of the Weaknesses, the only contribution of this paper is the TOC. the authors describe in Section Introduction that one of the innovations of this paper is that the TOC does not require additional training, however, in Section 3.2, the authors do give the need for the TOC to solve the optimization problem Equation (13).

**Questions:**

1. How to guarantee that the extracted fine modal-specific features are favorable to the downstream task rather than noise and background semantics?

2. How to optimize the Equation (13)?

3. In lines 152-153, what is the superscript $n$ in $f_i^n$?

4. The motivation for TOC being proposed is very subjective. From the explanations given by the authors, the TOC does not guarantee that the excluded features are irrelevant to the downstream task. Therefore, the authors need to give a theoretical analysis or a more reasonable and intuitive explanation that TOC can work. Otherwise the TOC will be much less innovative.

---

### Note · Authors · 2024-12-06

I have read and agree with the venue's withdrawal policy on behalf of myself and my co-authors.